# Model Study for Interaction of Sublethal Doses of Zinc Oxide Nanoparticles with Environmentally Beneficial Bacteria *Bacillus thuringiensis* and *Bacillus megaterium*

**DOI:** 10.3390/ijms231911820

**Published:** 2022-10-05

**Authors:** Katarzyna Matyszczuk, Anna Krzepiłko

**Affiliations:** Faculty of Food Sciences and Biotechnology, University of Life Sciences in Lublin, St. Skromna 8, 20-704 Lublin, Poland

**Keywords:** *Bacillus*, ZnO nanoparticles, antibacterial activity, ROS, biofilm, IAA, Evans blue decolorization

## Abstract

Zinc oxide nanoparticles (ZnO NPs), due to their antibacterial effects, are commonly used in various branches of the economy and can affect rhizobacteria that promote plant growth. We describe the effect of ZnO NPs on two model bacteria strains, *B. thuringiensis* and *B. megaterium*, that play an important role in the environment. The MIC (minimum inhibitory concentration) value determined after 48 h of incubation with ZnO NPs was more than 1.6 mg/mL for both strains tested, while the MBC (minimum bactericidal concentration) was above 1.8 mg/mL. We tested the effect of ZnO NPs at concentrations below the MIC (0.8 mg/mL, 0.4 mg/mL and 0.2 mg/mL (equal to 50%, 25% and 12,5% MIC, respectively) in order to identify the mechanisms activated by *Bacillus* species in the presence of these nanoparticles. ZnO NPs in sublethal concentrations inhibited planktonic cell growth, stimulated endospore formation and reduced decolorization of Evans blue. The addition of ZnO NPs caused oxidative stress, measured using nitroblue tetrazolium (NBT), and reduced the activity of catalase. It was confirmed that zinc oxide nanoparticles in sublethal concentrations change metabolic processes in *Bacillus* bacteria that are important for their effects on the environment. *B. thuringiensis* after treatment with ZnO NPs decreased indole acetic acid (IAA) production and increased biofilm formation, whereas *B. megaterium* decreased IAA production but, inversely, increased biofilm formation. Comparison of different *Bacillus* species in a single experiment made it possible to better understand the mechanisms of toxicity of zinc oxide nanoparticles and the individual reactions of closely related bacterial species.

## 1. Introduction

Zinc oxide nanoparticles (ZnO NPs) are among the most commonly used nanoparticles. The estimated global annual production of ZnO NPs in 2010 was >30,000 tones [1]. Due to their unique chemical–physical properties, these nanoparticles are widely used for commercial and industrial purposes. Zinc oxide nanoparticles (ZnO NPs) are an important product used in various branches of industry due to their optical, catalytic and electrical properties [2,3]. They are used in the food industry, agriculture and medicine as a UV filter and as a carrier for targeted drug delivery [2,3,4,5,6]. In agriculture, ZnO NPs are a component of fertilizers to support plant development and as nanopesticides [6]. Moreover, numerous studies have reported the antimicrobial efficiency of ZnO NPs against a broad-spectrum of pathogens, so these nanoparticles can be used as an antimicrobial agent [5,6]. Many examples of toxic effects of nanoparticles on various bacteria species have been described [5,6,7]. Evidence has indicated that ZnO NPs have a stronger antibacterial effect on Gram-positive bacteria than on Gram-negative bacteria [8,9].

The widespread use of engineered nanoparticles may entail the risk of increasing their concentrations in ecosystems and uncontrolled effects on the environment. In 2010, estimated emissions of ZnO NPs were 90–578 tons/year to the atmosphere, 170–2985 tons/year to water bodies and 3100–9283 tons/year to soils [1]. Model studies suggest that among various elements of the environment, soil microbes are particularly vulnerable to the effects of nanoparticles because soil can absorb the nanoparticles, causing long-term effects and locally increasing concentrations of toxic substances [9]. Bacteria are one of the first targets of nanoparticles in the environment [5,6,10]. The widespread use of zinc oxide nanoparticles prompted us to investigate the effect of sublethal concentrations of zinc oxide nanoparticles on the ecosystem-important plant growth-promoting rhizobacteria (PGPR) commonly occurring in soil [9]. The species, belonging to the genus *Bacillus*, are a common PGPR in soil that play a key role in conferring biotic and abiotic stress tolerance to plants by induced systemic resistance [11]. They are capable of forming endospores and are primarily saprophytes that break down organic compounds, and thus are able to colonize a variety of environments and contribute to nutrient cycling. *Bacillus* species produce various enzymes, antibiotics, vitamins, amino acids, organic acids and other substances, performing important functions in the natural environment as well as in the biotechnological industry [11,12,13]. Various *Bacillus* species have the potential to support plant growth, limit or fight plant pathogens and mitigate the effects of stress, e.g., salt stress, by colonizing the roots and regulating physiological and molecular processes in plants [11,12,13,14,15,16]. *Bacillus* sp. colonize the soil environment and directly influence soil quality and the condition of plants, as they are involved in processes such as nutrient cycling, organic matter decomposition and symbiotic relations with plants [15]. Some *Bacillus* species benefit plants by supplying them with minerals via reduction of atmospheric nitrogen to forms available to plants. The use of *Bacillus* strains in crop cultivation beneficially affects plants, enhancing phosphorus uptake and plant growth [12]. Numerous studies have shown that many *Bacillus* species are able to synthesize auxins [13]. *Bacillus* species, by secreting organic acids into the soil (e.g., gluconic, lactic, acetic, succinic and propionic acid), dissolve phosphates and increase soil fertility. Certain *Bacillus* strains can induce systemic resistance in plants and increase their tolerance to adverse environmental factors [14,15,16]. In nature, *Bacilli* are well known to be capable of degrading and transforming various chemicals. Some *Bacillus* bacteria have been shown to play an important role in bioremediation and mineralization due to their ability to remove azo dyes [17]. They form the backbone of many aquatic and terrestrial food webs, and therefore, assessment of the effects of NPs on various *Bacillus* species is crucial to understanding their reactions and predicting environmental consequences.

Given the importance of *Bacillus* to ecosystems, the interactions of NPs–PGPR are crucial. The microbial toxicities of zinc oxide nanoparticles were evaluated for two model bacteria strains: *B. thuringiensis* and *B. megaterium*. Like other members of the genus *Bacillus*, these strains also play an important role in the environment and support plant development [18,19,20]. The model organisms used in this study were incubated with ZnO NPs as a stress factor, applied at concentrations of 50%, 25% and 12,5% of the MIC (minimum concentration in mg/mL that inhibits the growth of microorganisms). Examination of the cells shows their reactions to stress caused by the presence of ZnO NPs and makes it possible to predict the environmental consequences by focusing on assessment of biofilm-forming ability, degradation of the azo dye Evans blue, production of indole acetic acid (IAA), growth capacity and spore formation. Comparison of these model *Bacillus* species in a single experiment also shows their individual responses to various concentrations of zinc oxide nanoparticles.

## 2. Results

### 2.1. Growth Parameters of Bacillus spp. after Incubation with ZnO NPs

The aim of the experiment was to determine the response of two model rhizobacteria, *B. thuringiensis* and *B. megaterium*, to sublethal doses of well-characterized engineered nanoparticles: ZnO NPs [21]. First, the antibacterial effects of nanoparticles were routinely assessed by MIC testing [22]. The MIC value determined was higher than 1.6 mg/mL, and MBC was higher than 1.8 mg/mL for both strains. *Bacillus thuringiensis* and *B. megaterium* are ubiquitously present in soil and water ecosystems, where they are known to exert a broad range of beneficial effects [11,12,13,14,15,16,17]. We therefore sought to assess the effects of ZnO NPs on *Bacilli* in order to elucidate the toxicity mechanisms involved and to gauge the potential impact of ZnO NPs exposure. We treated *B. thuringiensis* and *B. megaterium* cultures with 0.8 mg/mL, 0.4 mg/mL and 0.2 mg/mL (equal to 50%, 25% and 12,5% MIC, respectively) ZnO NPs and evaluated bacterial growth and metabolism.

Planktonic growth of cells in the presence of ZnO NPs sublethal doses was monitored in liquid medium in the stationary phase after 48 h by measuring the OD of the culture. The planktonic growth of *B. thuringiensis* and *B. megaterium* cells in the control were similar. The addition of nanoparticles delayed the planktonic growth of cells. The reactions of *Bacillus* strains to the presence of zinc nanoparticles in the growth environment were similar: as the concentration of nanoparticles increased, the optical density of the culture decreased (Figure 1a). At the ZnO NPs concentration of 0.8 mg/mL, the optical densities of the cultures of *B. thuringiensis* and *B. megaterium* were 84% and 48% of the control value, respectively, which may suggest greater sensitivity of *B. megaterium* (Table 1). However, statistical analysis (ANOVA) confirmed that the concentration of ZnO NPs influences the OD value, but the type of strain does not affect the OD value of a culture treated with ZnO NPs.

One of the most primary survival mechanisms for bacteria is forming spores to protect themselves against environmental degrading agents. Spore formation is also a common survival mechanism observed in *Bacillus* spp. Schafer–Fulton staining was used to determine the percentage of endospore-producing cells [23]. In the cultures of *Bacillus* species. the number of spores were very similar (Figure 1b, Table 1). In the control cultures of both *Bacillus* strains, after 48 h, the percentage of spores was about 15%. The percentage of sporulating cells increased with ZnO NPs concentration. 

### 2.2. Oxidative Stress in Bacillus Cells in the Presence of ZnO NPs

Oxidative stress was assessed using NBT and by determining catalase activity. Generation of superoxide anion radical in *Bacillus* cells was determined by measuring the reduction of NBT to formazan [24]. Production of superoxide anion radicals was low in the control samples of both *Bacillus* strains, as the absorbance of the solution was low, indicating the presence of formazan (Figure 2a). It should be noted that the amount of superoxide anion produced in the control *B. megaterium* was 30% lower than in *B. thuringensis*. Production of this reactive oxygen species increased in the presence of ZnO NPs depending on the concentration of nanoparticles in both analyzed strains. The experiment showed varied responses of the *Bacillus* strains to stress induced by ZnO NPs. *B. thuringiensis* exhibited the highest stress level (210% and 297% of the control values) following treatment with ZnO NPs at 0.4 mg/mL and 0.8 mg/mL, respectively. *B. megaterium* produced less superoxide anion radical at all concentrations of ZnO NPs. Formazan absorbance was maximally increased by 50%. However, compared to the second strain, at 0.8 mg/mL doses (50% MIC), it was as low as the control value of *B. thuringiensis*. Furthermore, higher correlation exists between superoxide production of *B. thuringiensis* and ZnO NPs concentrations, as shown in Table 1.

Catalase is a key antioxidant enzyme and plays an important role in assessment of the toxic effect of compounds on living organisms [25]. Changes in its activity are an indication of oxidative stress, and thus of the harmfulness of a chemical compound. Higher catalase activity in the control samples was noted in *B. thuringiensis*. The presence of ZnO NPs in the growth environment of cells caused a decrease in catalase activity (Figure 2b). For both strains tested, a concentration of just 0.2 mg/mL decreased the activity of this enzyme by about 30%. A concentration of 0.8 mg/mL reduces the catalase activity by 57% in *B. thuringiensis* and by 64% in *B. megaterium*.

### 2.3. Effect of ZnO NPs on Biofilm Formation

The ability of bacteria to form a biofilm in the presence of various concentrations of ZnO NPs was determined using the crystal violet assay, which is commonly used for biofilm evaluation [26]. This dye binds to bacterial cells and components of the extracellular matrix with negatively charged molecules, such as polysaccharides and eDNA [26,27]. The OD values representing the quantity of biofilm in the control cultures of *Bacillus* after 48 h of incubation were varied (Figure 3). The highest value was noted for *B. thuringiensis* (OD = 0.44), while *B. megaterium* formed a biofilm with lower intensity (OD = 0.27). The biofilm formation reaction in the presence of ZnO NPs was varied for both *Bacillus* species (Figure 3, Table 1). A negative correlation exists between biofilm production of *B. thuringiensis* and ZnO NPs concentration (Table 1). On the contrary, a positive correlation was observed for *B. megaterium*. For *B. thuringiensis*, the predominant reaction to the presence of nanoparticles was the inhibition of biofilm formation, while *B. megaterium* were stimulated to form a biofilm. 

### 2.4. IAA Production

The capacity of *Bacillus* species to produce IAA was tested using Salkowski reagent [28]. In general, IAA was produced in both the control conditions and in the samples treated with ZnO NPs (Figure 4). In the control sample of *B. megaterium*, a concentration of IAA of 32 µg/mL was measured. A similar ability to produce IAA was recorded for *B. thuringiensis*—30 µg/mL IAA. The production of this phytohormone by the analyzed bacterial species was influenced by the concentration of ZnO NPs. The IAA level in the presence of ZnO NPs was bigger in *B. thuringiensis* than the levels detected in the control medium, whereas *B. megaterium* showed a decrease in IAA production, especially at a concentration of 0.8 mg/mL ZnO NPs. The results revealed a strong negative correlation between ZnO NPs concentration and IAA production in *B. megaterium*. However, a positive correlation was observed for *B. thuringiensis* (Table 1).

### 2.5. Decolorization of the Azo Dye Evans Blue

Higher concentration of Evans blue could affect microorganism growth [29]. In our study, a concentration of 0.1 mg/mL Evans blue was used and did not affect cell growth in the control culture. *Bacillus* strains are capable of decolorizing Evans blue (Figure 5). In our experiment, intensive degradation of the dye was noted in the control samples of *B. thuringiensis* (72.6%). *Bacillus megaterium* degraded about 41% of the initial amount of Evans blue. Doses of ZnO NPs were also shown to have intensive effect on the ability of *Bacillus* strains to decolorize the dye (Table 1). In *B. thuringiensis*, nanoparticles strongly inhibited the decolorization process; even at the lowest concentration of 0.2 mg/mL ZnO NPs, decolorization of Evans blue was reduced to 29.4%, while higher concentrations of nanoparticles reduced decolorization less intensively (at 0.8 mg/mL ZnO NPs it was reduced to 22%). In the case of *B. megaterium*, the decolorization was less intensive in the control samples. The addition of nanoparticles did not inhibit this process as strongly as in *B. thuringiensis*, as 35% decolorization was observed at the concentration of 0.2 mg/mL ZnO NPs. The difference between the control decolorization and the decolorization at the highest 0.8 mg/mL ZnO NPs concentration was 14% in *B. megaterium*, while in *B. thuringiensis* it was 50%.

## 3. Discussion

The study of model organisms is a key element in our understanding of the risks posed by the widespread use of engineered nanoparticles. Both *B. thuringiensis and B. megaterium* are known as a model in microbiological, biochemical, genetic and environmental research, especially related to the promotion of plant growth [12,13,14,16]. We used engineered ZnO NPs in the study due to their broad applications in various areas of biological science [2,3,4,5,6]. Many studies have been conducted to demonstrate the antimicrobial activity of zinc oxide nanoparticles against various microorganisms [2,4,5,6]. The effect of zinc oxide nanoparticles on this genus of bacteria has been determined to a very limited degree, usually with respect to *B. subtilis* and in a narrow range of concentrations [7,22,30,31,32,33]. Some authors described the antimicrobial effect of nanoparticles obtained by the green synthesis method. In our experiment we used commercial nanoparticles of zinc oxide because they can pollute the environment on a large scale. For both *Bacillus* strains, we obtained the same values of MIC > 1.6 mg/mL and MBC > 1.8 mg/mL. To our knowledge, this is the first report of the MIC for engineered ZnO NPs for *B. megaterium*. Literature reports give varied MIC/MBC values for zinc nanoparticles for various bacteria, with MIC ranging from 80 µg/mL to over 3000 µg/mL, and MBC from 150 µg/mL to over 3000 µg/mL [22,30,31,32,33]. For representatives of *Bacillus* as well, the literature reports various values: MBC for *B. subtilis* of 12 μg/mL [30], MIC for *B. subtilis* MTCC of 441 and *Bacillus* sp. clsxc_TYA (MH884600) of 0.46 mg/mL and 0.51 mg/mL, respectively [22], MIC for *B. subtilis* of 0.105 mg/mL, MIC for *B. thuringiensis* of 0.120 mg/mL [31], and MIC for *B. thuringiensis* of 1 mg/mL and MBC of 1.5 mg/mL [33]. Many of these studies suggest that the degree of antibacterial effects of zinc oxide nanoparticles is proportional to their concentration, but their size, shape and charge, as well as stabilizing agents, also play an important role [34]. Lower MIC values are usually obtained when the ZnO NPs tested are synthesized by the researchers themselves using various chemical and biological methods. The reactivity and biocidal properties of nanoparticles obtained in this manner may be influenced by residues of the reductant and stabilizer and byproducts of the synthesis process. The presence of additional chemical substances affects the biocidal properties of zinc oxide nanoparticles. This has been confirmed in a study on the effect of ZnO NPs on *B. subtilis* in the natural aqueous environment of two rivers, where, depending, on the origin of the water, there was either a significant reduction in the viability of bacteria or no effect on their survival [7]. Many factors influence the toxicity of ZnO NPs for bacteria: the growth medium, the method used to determine the survival rate and the incubation time. In order to evaluate sublethal ZnO NPs effects on two important representatives of rhizobacteria, namely *B. thuringiensis* and *B. megaterium*, we performed growth inhibition tests in liquid media. Our experiment confirmed that the growth rate, measured as the OD of the *Bacillus* culture, decreased as the nanoparticle concentration increased (Figure 1a). Generally, the literature data suggests that the antibacterial activity of ZnO NPs is concentration-dependent, but zinc oxide nanoparticles, especially at low concentrations, can stimulate the growth of some species of bacteria [32,35]. In *Azotobacter chroococcum* W5 growing in a medium with ZnO NPs, the optic density of the culture was shown to increase with nanoparticle concentration [35]. Bacteria use Zn ions as a cofactor in metabolic processes, and they have a beneficial effect on enzymes such as dehydrogenase, thiol peroxidase and glutathione reductase [32]. In our experiment, we observed a decrease in the growth rate (Figure 1a) as well as a decrease in the activity of the key antioxidant enzyme catalase (Figure 2b).

One of the most commonly studied mechanisms of the antibacterial action of ZnO NPs is the generation of reactive oxygen species, which damage cells and induce oxidative stress [36]. Chemical compounds present on the cell surface (carboxyl, phosphate and amino groups) give it a negative charge, which is conducive to the accumulation of ZnO nanoparticles on the cell surface and their adsorption in the peptidoglycan layer. ROS generation in a bacterial culture can take place in the dark [37]. Free radicals are generated on the surface of ZnO nanoparticles due to interactions with oxidants present in the reaction environment; furthermore, their electrochemical properties are conducive to the formation of reactive groups. In these reactive sites, an electron donor or acceptor interacts with molecular oxygen to form superoxide anion (O_2_^−^), which in turn can generate more ROS [36]. Depending on the type of stress factor, severe effects in microorganisms can be observed just half an hour after the stressor is applied [38]. Our experiments confirmed that the presence of sublethal doses of ZnO NPs in a bacterial culture increased production of superoxide anion radical (Figure 2a), depending on the concentration of nanoparticles, and reduced catalase activity (Figure 2b), which is indicative of oxidative stress in *Bacillus* cells. Scavengers of reactive oxygen species (ROS), such as catalase, are ubiquitous in nature and are known to play a key role in adaptation to stress and the survival of various species of microbes. *B. cereus*, closely related to our strains, demonstrates resistance to mild stress through changes in catalase activity [39]. Metal nanoparticles induced a significant increase in ROS and toxic effects associated with oxidative stress in *B. subtilis* [40]. Catalase plays an important role in protecting cells against the toxic effects of photoreactive nanoparticles of titanium oxide, with catalase-positive bacteria found to be more resistant than catalase-negative bacteria [25,41]. The reduction in catalase activity observed in our experiments under the influence of sublethal doses of zinc oxide nanoparticles may indicate a strong reaction of the cell to oxidative stress.

In response to the presence of zinc oxide nanoparticles, *Bacillus* cells activate defense mechanisms, including production of spores (Figure 1b) and growth of biofilm structures (Figure 3). In our experiment, a manifestation of the defense response may have been the increased production of spores by cells incubated with zinc oxide nanoparticles (Figure 1b). According to the literature data, ZnO NPs can inhibit sporulation and significantly modify metabolism in *B. subtilis* under long-term adaptation growth conditions, but 48 h incubation with ZnO NPs increased sporulation [10].

Many species of gram-positive and gram-negative bacteria are able to form a biofilm, which contains attached bacterial cells enclosed in an extracellular matrix. Biofilms are created in natural conditions, e.g., on plant roots, and are also the response of bacteria to toxic substances appearing in their environment, protecting them against both unfavorable environmental conditions and antibacterial agents [26,27,33,40]. The *Bacillus* species tested in our study form biofilms (Figure 3). Biofilm formation can be limited by metal nanoparticles, especially at high concentrations. Ag NPs have been shown to inhibit biofilm formation by *Pseudomonas aeruginosa* [42], *E. coli* and *Streptococcus mutans* [43]. Inhibition of biofilm formation under the influence of Cu NPs has also been observed in *Pseudomonas aeruginosa* [44] and *Listeria monocytogenes* [45]. Gram-negative bacteria are more sensitive to Ag NPs than gram-positive bacteria are [43]. Our experiment showed varied biofilm formation reactions in bacteria in the presence of ZnO NPs (Figure 3). Sublethal concentrations of ZnO NPs were shown to stimulate biofilm formation in *B. megaterium*, but inhibited it in *B. thuringiensis*. These results suggest that peptidoglycans secreted extracellularly can be a barrier for nanoparticles [35,46,47]. Many authors, however, stress that zinc oxide nanoparticles can reduce bacterial biofilm formation. Zinc oxide nanoparticles reduce biofilm-forming capacity and cause the hydrophobicity index to fall in cultures of the gram-negative bacteria *P. aeruginosa* and *S. aureus* [46]. ZnO NPs in a nanocomposite with nanosilver and chitosan exert an anti-biofilm effect against gram-positive bacteria, such as *Bacillus licheniformis* and *Bacillus cereus*, and gram-negative bacteria, such as *Vibrio parahaemolyticus* and *Proteus vulgaris* [47].

Zinc oxide nanoparticles affect metabolic processes in *Bacillus* that are important for the effects of these bacteria on the environment [13]. One of the manifestations of the potentially negative impact of ZnO NPs is their effect on production of IAA. Indole-3-acetic acid (IAA) is a natural auxin produced by bacteria, as well as by fungi and plants. The positive effect of rhizosphere bacteria (which include the genus *Bacillus*) on plants is linked to the production of this plant growth hormone, which supports the growth and development of plants [13,28]. In our study, both *Bacillus* species produced IAA both in the control medium and the media with nanoparticles (Figure 4). Our studies show different IAA production in *B. thuringiensis* and *B. megaterium* in response to sublethal doses of ZnO NPs. The level of IAA produced by *B. megaterium* in the presence of ZnO NPs was much lower than that in the control media. On the other hand, in *B. thuringiensis*, IAA production increased as the concentration of sublethal doses of ZnO NPs increased. More research is still needed to use this effect in plant growth stimulants. The widespread use of ZnO NPs increases the risk of their harmful effects on the environment, including soil [9]. One of the potential environmental effects is impairment of the ability of bacteria to synthesize IAA. Similar results to ours for *B. megaterium* have been obtained in *Pseudomonas aeruginosa*, *P. fluorescens* and *B. amyloliquefaciens*, as IAA production fell as the concentration of ZnO NPs and TiO_2_ NPs increased, with ZnO NPs exerting a more harmful effect than TiO_2_ nanoparticles [47]. Dimkpa et al. found that the addition of CuO NPs and ZnO NPs to the medium modified IAA levels relative to the control in Gram-negative *P. chlororaphis* [48]. They also observed that CuO NPs significantly increased IAA production in a 48 h culture, whereas ZnO NPs decreased the IAA level relative to the control. Zinc oxide nanoparticles also affect the ability of *A. chroococcum* W5 to produce IAA. In the presence of a suspension of ZnO NPs (50 mmol/L), IAA production fell by 80% relative to the control [35].

Another effect of the toxicity of ZnO NPs for *Bacillus* in our experiment was a decrease in decolorization of Evans blue dye (Figure 5). This synthetic organic dye contains azo groups associated with aromatic rings and is resistant to the effects of physical and chemical factors. Due to its widespread occurrence in wastewater, Evans blue poses a threat to aquatic ecosystems and to entire food webs associated with them [29]. Some microbes are capable of biotransformation of azo dyes, including Evans blue—mainly bacteria of the genera *Pseudomonas, Bacillus, Sphingomonas, Aeromonas, Citrobacter, Escherichia, Desulfovibrio, Proteus, Shewanella* and *Alcaligenes* [49]. In the genus *Bacillus*, strains with a high capacity to degrade various azo dyes have been isolated [29,48]; this ability has been detected in a selected strain of *B. cereus* [50] and in a mixed culture of *B. vallismortis, B. pumilus, B. cereus, B. subtilis* and *B. megaterium* [51]. Some *Bacillus* strains are able to degrade azo dyes in a wide range of concentrations up to 500 mg/1 [21,52]. The authors of these studies emphasize that decolorization of azo dyes by *Bacillus* spp. takes place through degradation of azo compounds and not absorption by the cells. It has also been established that this dye at high concentrations can inhibit the growth of bacterial cells [49]. The *Bacillus* strains were not selected for degradation of azo dyes, but both strains in the control culture caused 73% or 41% decolorization of the initial amount of dye, with the most effective being *B. thuringiensis*. The presence of ZnO NPs and Evans blue dye in the growth environment of bacteria reduced the degree of decolorization of the dye in all cases, but the reactions of individual species to increasing concentrations of ZnO NPs in the environment were varied. Rapid decolorization of azo dyes by microbes can be increased using redox mediators, which accelerate electron transfer from the donor to the acceptor. This role could be played by metal nanoparticles; this has been confirmed for iron oxide nanoparticles. Decolorization and degradation of methyl red by immobilized cells of *Aeromonas jandaei* SCS5 in anaerobic and aerobic conditions increased in the presence of magnetic Fe_3_O_4_ nanoparticles compared to inactivated cells alone [53]. Similarly, high efficiency of decolorization of Basic Red 46 azo dye was obtained using *Enterobacter cloacae* coated with synthesized magnetic Fe_3_O_4_ NPs [54]. Iron compounds, due to their high positive redox potential (Fe^3+^ + e^−^ → Fe^2+^; E_0_ = + 0.77), can enter into redox reactions more easily than can zinc compounds (Zn^2+^ + 2 e^−^ = Zn; E_0_ = −0.76) and therefore accelerate the degradation of azo dyes. As mentioned in the ‘Materials and Methods’, decolorization of Evans blue did not take place in the growth medium with dye and various concentrations of ZnO NPs but without bacteria. In UV-irradiated samples, zinc oxide nanoparticles can increase the rate of degradation of azo compounds through photocatalysis [55], but in our experiment, a UV lamp was not used; the cultures were incubated in an incubator (in the dark). The decrease in decolorization of Evans blue in our study was therefore caused by the effect of zinc oxide nanoparticles on *Bacillus* cells and the reduction in metabolic processes leading to biotransformation of Evans blue.

Model organisms such as *B. thuringiensis* and *B. megaterium* can help understand the effects of sublethal doses of ZnO NPs on rhizosphere bacteria and their role in the environment. The increasingly common use of ZnO NPs in agriculture may disrupt the metabolism of *Bacillus* bacteria and reduce their environmental role as bacteria promoting plant growth. Statistical analysis showed that ZnO NPs have significant effects on the analyzed parameters of growth and metabolic activity, except for catalase (Table 1). ANOVA showed that the type of strain did not statistically significantly influence the OD value, the number of endospores or the catalase activity in the cultures treated with ZnO NPs. In the cultures treated with ZnO NPs, the type of strain statistically significantly influenced biofilm formation, IAA production, production of superoxide anion radical and decolorization of Evans blue. 

## 4. Materials and Methods

### 4.1. Zinc Oxide Nanoparticles

Zinc oxide nanoparticles (ZnO NPs) were purchased from Sigma Aldrich Germany (catalog no. 721077). The ZnO NP dispersion was synthesized by hydrolysis of zinc salt in a polyol medium heated to 160 °C. This product has a reported particle size <100 nm as measured by dynamic light scatting (DLS) and an average particle size <35 nm as measured using an aerodynamic particle sizer (APS) spectrometer. Wang et al. [21] analyzed this preparation using a Zetasizer Nano (Malvern Instruments, Worcestershire, UK) and obtained an average weighted particles size of 67 ± 2 nm and with a zeta potential of + 46.1 ± 1.5 mV. 

### 4.2. Bacteria and Determination of Growth Parameters

Bacillus strains were purchased from the Polish Collection of Microorganisms (PCM) Wroclaw, Poland and were registered in the World Federation for Culture Collections (WFCC, no. 106) and the European Culture Collections’ Organisation (ECCO). The PCM possesses the status of international depository authority (IDA) for patent purposes. In this study, *B. thuringiensis* PCM 1853 and *B. megaterium* PCM 2023 were used as model rhizosphere bacteria. Strains were maintained at −80 °C in suspensions containing 20% glycerol. 

MIC (the minimum concentration in mg/mL that inhibits the growth of microorganisms, measured as OD) was determined by the microdilution method [22]. ZnO NPs solutions (from 1.4 μg/mL to 2 mg/mL) in nutrient broth were inoculated with bacteria (for a final concentration of 5 × 10^4^ CFU/mL), applied to the wells of a BioScreen C microplate (Bioscreen C, Labsystems, Helsinki, Finland), and incubated at 37 °C with shaking for 24 h. Control samples were also added to the wells: a control of the growth of each of the bacteria (medium + bacterial inoculum) and a control of the turbidity of the solution (medium + ZnO NPs at each concentration without bacteria). The optical density was measured at 600 nm and automatically recorded every 30 min with the Bioscreen C microbiology reader. The MIC was taken as the lowest concentration of ZnO NPs at which the growth of the bacteria was completely inhibited after 24 h of incubation at 37 °C. MBCs were determined by agar plate method. Contents from wells that did not show growth and from the control were transferred to solid nutrient agar without nanoparticles and checked for growth. The mean numbers of colonies in the control samples were 7.5 × 10^8^ CFU/mL for *B. thuringiensis* and 8.6 × 10^8^ CFU/mL for *B. megaterium*. MBC endpoint is defined as the lowest concentration of antimicrobial agent that kills 99.9% of the initial bacterial population.

To assess the response of bacteria to sublethal concentrations of nanoparticles, we used concentrations of ZnO NPs of 0.8 mg/mL, 0.4 mg/mL and 0.2 mg/mL (equal to 50%, 25% and 12.5% MIC, respectively). Liquid nutrient broth containing various dilutions of ZnO NPs was inoculated with a 24 h inoculum of bacteria to a final concentration of 5 × 10^4^ CFU/mL of medium and incubated at 37 °C for 48 h. The resulting cultures were used for tests based on NBT, catalase activity, endospore formation, degradation of Evans blue and biofilm assay by crystal violet. Control experiments were carried out without ZnO NPs.

Endospore staining was with the Schaeffer–Fulton method using malachite green and safranin [23]. All cells and endospores were counted under a light microscope, the number of endospores was expressed as a percentage.

### 4.3. Determination of Superoxide Radicals in Stress Conditions Induced by ZnO NPs

The concentration of superoxide radicals was determined by spectrophotometry with nitroblue tetrazolium (NBT). In an alkaline environment, superoxide anions cause NBT to form formazan [24]. The reaction mixture was prepared from 3 mL of distilled water, 0.05 mL of 1 M NaOH, 0.1 mL of 5 mM NBT solution, and 0.2 mL of bacterial culture incubated with ZnO NPs or a control sample and stored in the dark for 30 min at 20 °C. The absorbance was measured at 560 nm.

### 4.4. Determination of Catalase Activity

Hydrogen peroxide has the ability to penetrate cells, where it is broken down to oxygen by catalase. Gaseous oxygen is released from the solution in a calibrated Eykman tube (Labor-Glass, Lublin, Poland) and its quantity can be measured in mL [25]. After 48 h of incubation, 5 mL of each culture (prepared as described above) was centrifuged and suspended in 1 mL saline and transferred to an Eykman tube. Bacteria were placed in the closed arm of the Eykman tube, and 25 mL of 3% H_2_O_2_ was added. The height of the column of gaseous oxygen was measured every minute. The unit of catalase activity (U) is the volume of oxygen in mL released in 1 min by 1 mL of suspension of bacterial culture with OD = 1.

### 4.5. Crystal Violet Assay to Determine the Effect of ZnO NPs on Biofilm Formation

Biofilm formation was tested by crystal violet assay [26]. A culture of *Bacillus* cells was prepared with various concentrations of ZnO NPs and placed in the wells of Bioscreen C microplates. After 48 h of culture, the supernatant was poured off, and the wells were washed twice with saline to remove cells that were not bound in the biofilm structure. The plates were air-dried for 20 min, and then 400 µL of 0.1% crystal violet was added to the wells and left to stain them for 20 min. The crystal violet was removed, the plates were rinsed thoroughly with pure water and dried for 30 min, and then 400 μL of 30% acetic acid was added to each well to re-dissolve the crystal violet, followed by incubation for 30 min at room temperature. The absorbance of the solution was measured by spectrophotometry at 600 nm [27]. The following were used as controls: (a) sterile medium and (b) medium with various concentrations of nanoparticles without bacterial cells. The wells with these controls were stained as well. Control (a) served as a blank sample, and control (b) was used to determine that ZnO NPs did not influence staining of the wells with crystal violet. The results were expressed as a percentage of the absorbance of a control sample of a given bacterial strain growing without nanoparticles.

### 4.6. IAA Production in Conditions of Incubation with ZnO NPs

Following sterilization, tryptophan-supplemented (100 μg/mL) nutrient broth with various concentrations of nanoparticles was inoculated with bacteria (OD = 0.5). The control cells and cells treated with ZnO NPs were incubated at 37 °C with shaking. After 48 h of incubation, 2 mL of culture from each experimental treatment was centrifuged at 10,000 rpm for 10 min. The supernatant was mixed with two or three drops of phosphoric acid (H_3_PO_4_) and 4 mL of Salkowski reagent (2% 0.5 M FeCl_3_ in 35% HClO_4_). The samples were incubated in the dark for 1 h. The absorbance in the supernatant was measured at λ = 530 nm, and IAA was quantified against a standard curve [28].

### 4.7. Degradation of Evans Blue

The azo dye Evans blue (C_34_H_24_N_6_Na_4_O_14_S_4_, Sigma-Aldrich) was used. The number of azo bonds in the molecule classifies it as a diazo dye. The molar mass of the dye is 960.81 g/mol. *Bacillus* spp. strains were tested in a test tube for decolorization of Evans blue on nutrient broth (glucose 20 g/L, peptone 5 g/L, yeast extract 2 g/L, MgSO_4_ 0.5 g/L, KH_2_PO_4_ 1 g/L and pH 6.8). The medium was inoculated with bacteria to OD = 0.5, and various concentrations of ZnO NPs (0.2 to 1.4 mg/mL) and a solution of Evans blue at a concentration of 0.1 mg/mL was added. The cultures were grown for 48 h, and the absorbances of the solutions were measured at λ = 606 nm. Additional control samples were prepared: (a) growth medium with dye and without bacteria was used to determine the initial absorbance A0 of the solution; (b) growth medium with dye and various concentrations of ZnO NPs without bacteria confirmed that incubation of the dye with nanoparticles does not decolorize it; (c) measurement of the OD of the bacterial culture on nutrient broth +0.1 mg/mL Evans blue after 48 h of growth showed that azo dye at a concentration of 0.1 mg/mL does not inhibit bacterial growth.

The degree of decolorization was determined from the following formula:Decolorization (%) = [(A_0_ − A_1_)/A_0_] × 100(1)
where A_0_ is the absorbance of the control sample of dye, and A_1_ is the absorbance of the sample after incubation with bacteria [29].

### 4.8. Statistics

Data for statistical computations were taken from at least three independent experiments, each performed in three replications. Statistical analysis was performed using analysis of variance (ANOVA) with 0.05 as the significance level. The correlation between variables was examined using the Pearson (r) linear correlation coefficient. The results are presented as means ± SD.

## 5. Conclusions

Our experiment confirms the multi-faceted mechanism of toxicity of zinc oxide nanoparticles for *B. thuringiensis* and *B. megaterium* strains, while underscoring the individual susceptibility of closely related bacterial species. We observed a different reaction of the strains with the presence of ZnO NPs—this concerned the production of IAA and the formation of a biofilm. The analyzed parameters of the responses of cells are indicative of strategies adapting bacteria to the presence of ZnO NPs. Such adaptations as intense spore formation may give bacteria an advantage in survival, but can also lead to ecological consequences, disturbances in the bacterial growth, biofilm formation, reactions for oxidative stress, promotion of plant growth and the biotransformation of toxins such as Evans blue. Demonstration of the response of the analyzed *Bacillus* species draws attention to the fact that the toxicity parameters of nanoparticles must be carefully selected. The responses of these strains (e.g., changes in IAA induced by ZnO NPs) may be helpful in evaluating environmental effects or in formulating preparations for use in agriculture.

## Figures and Tables

**Figure 1 ijms-23-11820-f001:**
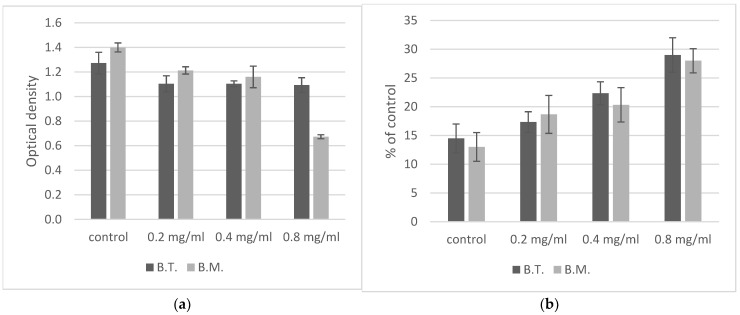
Concentration-dependent inhibition of *Bacillus* sp. planktonic cell growth by ZnO NPs. (**a**) Effects of zinc oxide nanoparticles on values of OD at 600 nm. Statistical tests confirmed that the concentration of ZnO NPs influences the OD value, but the type of strain does not affect the OD value of a culture treated with ZnO NPs. (**b**) Influence of nanoparticles on cell’s sporulation. The concentration of ZnO NPs statistically significantly affects the number of spores produced, but no statistically significant differences were shown between strains of bacteria. B.T.—*Bacillus thuringiensis*, B.M.—*Bacillus megaterium*.

**Figure 2 ijms-23-11820-f002:**
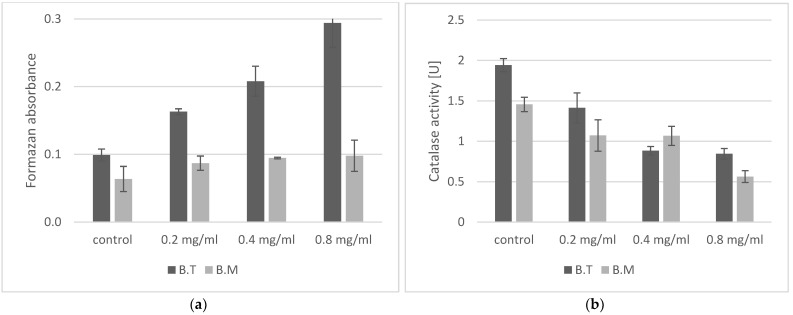
Parameters of oxidative stress induced by the presence of ZnO NPs in bacterial cells. (**a**) Changes to superoxide anions production in *Bacillus* sp. treated with ZnO NPs. ANOVA confirmed that the concentration of ZnO NPs influences the amount of formazan produced from NBT, and that the type of strain influences the amount of formazan produced in the presence of ZnO NPs. (**b**) Effect of different concentrations of ZnO NPs on catalase activity. The concentration of ZnO NPs does not statistically significantly affect catalase activity, and the type of strain does not statistically significantly affect catalase activity in the presence of ZnO NPs. B.T.—*Bacillus thuringiensis*, B.M.—*Bacillus megaterium*.

**Figure 3 ijms-23-11820-f003:**
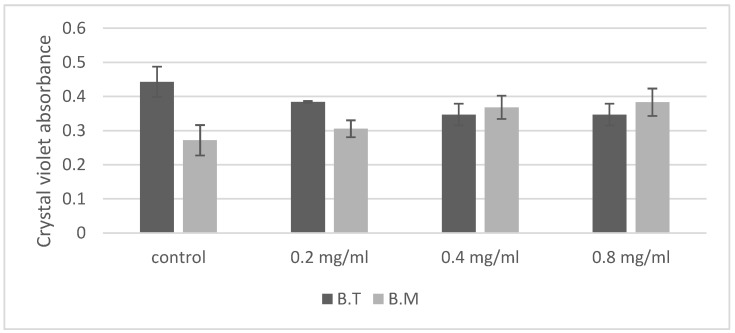
Effect of ZnO NPs on biofilm formation of *Bacillus* sp., expressed as crystal violet absorbance. The concentration of ZnO NPs statistically significantly affects the absorbance value of crystal violet, and the type of strain influences the absorbance of crystal violet in the presence of ZnO NPs. B.T.—*Bacillus thuringiensis*, B.M.—*Bacillus megaterium*.

**Figure 4 ijms-23-11820-f004:**
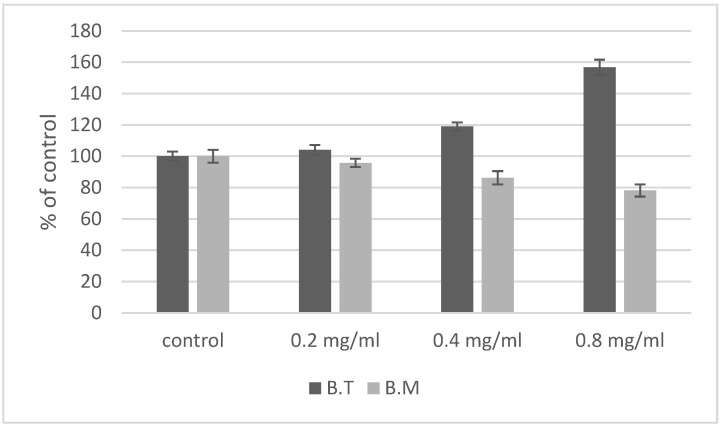
Effect of ZnO NPs on IAA production by cells of *Bacillus* sp. expressed as percent of the control value. The concentration of ZnO NPs statistically significantly affects the amount of IAA produced, and the type of strain influences the amount of IAA produced in the presence of ZnO NPs. B.T.—*Bacillus thuringiensis*, B.M.—*Bacillus megaterium*.

**Figure 5 ijms-23-11820-f005:**
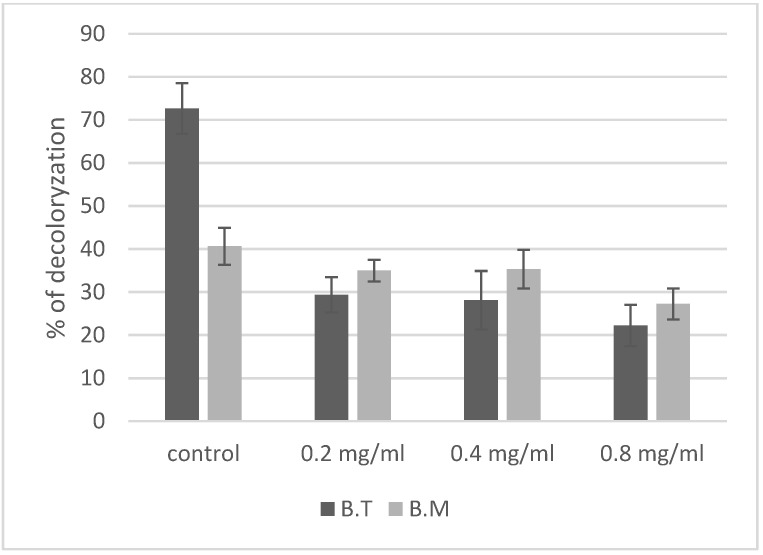
Effect of ZnO NPs on decolorization of Evans blue by cells of *Bacillus* sp. The concentration of ZnO NPs statistically significantly influences decolorization of Evans blue, and the type of strain influences its decolorization in the presence of ZnO NPs. B.T.—*Bacillus thuringiensis*, B.M.—*Bacillus megaterium*.

**Table 1 ijms-23-11820-t001:** Pearson correlation coefficients and ANOVA showing the effect of ZnO NPs on selected parameters of the growth and metabolic activity of *Bacillus* spp. strains.

Growth/ Metabolic Parameters	Pearson Correlation Coefficients		ZnO NP Dose Effect		Influence of Strain Type on ZnO NP Dose Response	
	B.T	B.M	*p*-Value	Test F	*p*-Value	Test F
Planktonic growth	−0.5929	−0.9488	3.9 × 10^−28^	2.748	0.118	3.991
Spore formation	0.9805	0.9696	7.0 × 10^−12^	3.239	0.142	4.494
NBT reduction	0.8341	0.5453	0.001	3.239	2.1 × 10^−5^	4.494
Catalase activity	−0.8816	−0.9597	0.731	4.757	0.069	5.153
Biofilm formation	−0.3723	0.7478	8.8 × 10^−37^	2.758	6.7 × 10^−7^	3.150
IAA production	0.6004	−0.9143	1.8 × 10^−5^	2.769	3.6 × 10^−21^	4.013
Evans blue decolorization	−0.7728	−0.8254	1.8 × 10^−12^	3238	3.9 × 10^−3^	4.499

Pearson correlation coefficients determined the effect of ZnO NPs on selected parameters of the growth and metabolic activity of strains: B.T—*Bacillus thuringiensis*, B.M.—*Bacillus megaterium*. Analysis of variance (ANOVA with 0.05 of significance level; *Df* = 3) determined *p*-value and Test F value.

## Data Availability

Not applicable.

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
