# Peer review of "Model Study for Interaction of Sublethal Doses of Zinc Oxide Nanoparticles with Environmentally Beneficial Bacteria Bacillus thuringiensis and Bacillus megaterium"

_ijms, 2022, doi:10.3390/ijms231911820_

Round 1

Reviewer 1 Report

The article 'Model study for interaction of sublethal doses of zinc oxide nanoparticles with environmentally beneficial bacteria Bacillus thuringiensis and Bacillus megaterium' is generally interesting and may be of interest to readers, but there is a lack of explanation as to why such results were obtained in the study, especially comparing results between one bacterium and another. I have submitted comments on the paper below.

1. Numerous double spaces in the text.

2. Explain abbreviations used e.g., in the abstract NBT, IAA

3. 90 -5 78 tons/year should be 90 - 578 tons/year?

4. Table 1 - two dots

5. No description of Figure 1B, why does an increase in ZnO NPs result in an increase in bacterial concentrations and for the first bacterium not? Please provide a literature reference.

6. The name ZnONPs is once written with a break (ZnO NPs) and once in lower case (ZnONps). Please unify this.

7. Introduction: Please add recent ZnO production data (at least from 2018), comparing with 2010.

8. Table 1. should be better described. How were the subsequent data, included in the table, determined/calculated? What do the correlations between parameters with concentration mean, the proportionality of the parameters with concentration was determined here?

9. Please change the title of Figure 2.

10. Materials and methods - there is limited information on how the experiments were performed, e.g. after what time was successive absorbance measured, were the experiments repeated three times, but as separate series, or was the series one and each solution bottled for three measurements?

11. It is necessary to compare the results with the positive control when determining MIC and MBC tests.

12. The characterisation of ZnO NPs is missing from the analysis. It is necessary to add SEM/TEM images (with appropriate zoom) to observe the shape of the particles, which is important for toxic properties. Additionally, zeta potential analysis is strongly recommended to determine the surface character of the nanoparticles.

13. „Literature reports give varied MIC/MBC values for zinc nanoparticles for various bacteria, with MIC ranging from 80 g/ml to over 3000 g/ml and MBC from 150 g/ml to over 3000 g/ml 230 [22, 30-33].” - Is it possible to disperse 3000 g ZnO in 1ml?

14. Please check your e-mail addresses, as there seems to be an error.

Author Response

Response to Reviewer 1 Comments

Thank you for your insightful comments on our manuscript: Model study for interaction of sublethal doses of zinc oxide nanoparticles with environmentally beneficial bacteria Bacillus thuringiensis and Bacillus megaterium.

Owing to your suggestions, the work will be easier to understand and we expect it to be well-received by its readers. In the revised version of the manuscript we have taken into account all of the comments concerning editing – removed the double spaces, explained the abbreviations, and corrected the spelling errors. We have also changed some of the units so that they are consistent, and we have added statistical information to the figure descriptions.

Below are our detailed responses to some of the reviewer’s comments.

Our study shows how two closely related species of bacteria react to sublethal doses of ZnO NPs. Both species can be components of formulations enhancing plant growth and are important species in the soil environment. Zinc oxide nanoparticles have wide application in fertilization and bioremediation and as antibacterial preparations. Demonstration of the response of the analysed Bacillus species draws attention to the fact that the toxicity parameters of nanoparticles must be carefully selected. The responses of these strains (e.g. changes in IAA induced by ZnO NPs) may be helpful in evaluating environmental effects or in formulating preparations for use in agriculture. Researchers most often focus on the toxic effects induced by nanoparticle concentrations equal to the MIC, while little is known of the effects of sublethal concentrations.

Suggestion 5. No description of Figure 1B, why does an increase in ZnO NPs result in an increase in bacterial concentrations and for the first bacterium not? Please provide a literature reference.

Authors: Fig. 1a illustrates the optical density of the culture in liquid medium, while Fig 1b presents the number of endospore-producing cells. We have added the results of the statistical analysis in Table 1 and Figure 1, and in the text of the manuscript lines: 405-411

Suggestion 7. Introduction: Please add recent ZnO production data (at least from 2018), comparing with 2010.

Answer: Unfortunately we were unable to find a reliable source of information concerninglobal annual production of ZnO NPs in recent years (including from 2018). Therefore we have left this literature reference in the manuscript: Keller, A. A.; Mc Ferran, S.; Lazareva, A.; Suh, S. Global life cycle releases of engineered nanomaterials. J. Nanoparticle Res. 2013, 15(6), 1-17.

Suggestion 8. Table 1. should be better described.

Authors: We have changed the title and description of Table 1:

Suggestion 9. Please change the title of Figure 2.

Authors: The new title of Fig. 2 is: Parameters of oxidative stress induced by the presence of ZnO NPs in bacterial cells.

Suggestion 10. Materials and methods - there is limited information on how the experiments were performed, e.g.

Authors: We have added information on how the experiments were conducted, describing measurement of MIV and MBC (lines 445-449, 453-455), as well as the statistical analysis (lines 528-532).

Suggestion 12. The characterisation of ZnO NPs is missing from the analysis.

Authors: We used nanoparticles manufactured commercially by Aldrich. Their characteristics are described by the manufacturer (lines 427-433). This information is given in the Material and Methods section.

Suggestion 13. ‘Literature reports give varied MIC/MBC values for zinc nanoparticles ‘

Authors: This was our mistake – we omitted the symbol [µ] (microgram). These concentrations are expressed in micrograms/ml.

Thank you for your time and insightful analysis of our manuscript.

Reviewer 2 Report

The revised manuscript studies the effects of sublethal doses of znic oxide nanoparticles on selected PGP traits of two Bacillus species. Authors estimated MIC and MBC what allowed them to select "safe" doses for their study. 

This is important topic as NPs can be used to promote some process and thus improve some measures. Authors indicated the fields of application of ZnO NPs what makes this manuscript valuable.

The manuscript is well organized with moderately good English. Please consider some its improvement. 

The title and keywords are good, abstract presents key information. Introduction is a brief description of NPs and their possible effects on soil microorganisms. The characteristics of Bacillus sp. are also presented. It ends with brief description of work done and its rationale (the aim).

Experiment is well designed and described allowing its repetition. I only miss statistical procedures as there are clear patterns which should be confirmed by statistical analysis (dose effects, differences between strains, differences to the control).

For example (Fig 2a): SOD generation was clear for BT but for BM the changes were rather insignificant - no ZnONPs effect on this trait for BM.

Presented correlations are not tested for their significance.

Results are clear, data are well described with good figures and tables. However, missing statistical analyses makes them less pronounced. Interestingly, conclusions mention the use of ANOVA but no methodology, description with data are provided. 

The statistical analyses would allow to finally state: (1) ZnONPs dose effect for both strains, (2) dose-dependent effects, and (3) show which strain (if any) displays better tolerance to NPs in terms of selected traits. As often can be seen, there are initial differences between strains (control). I strongly recommend to add information of these effects using statistical tools to each analyzed trait.

I would also provide IAA values for each treatment for better comparison with data presented in discussion.

Discussion is good using recent literature data, it ends with conclusions. However, conclusions need to be re-written taking into account the results of statistics and respond to the aim more in detail.

Please check the whole text for typing issues, e.g.:

- additional spaces (for example title starts with one space)

- line 33 - there is a missing space in "was> 30,000 tones"

- L38: "In the field of agriculture. ZnONP is" - note there is a dot

- L55, 92, 198: please italize Bacillus, L103 - the same for Bacilli, also in Fig. 1 caption

- the titles of subsections in methods should be written according to the Authors Guide

- what was the origin of stains used?

- L408: there is missing upper index in 5x104 CFU/ml, the same in line 421

- please add a comma before 'respectively in line 420

- L438, 440, 443: cm3 misses upper index. As in other places Authors use 'ml' consider also use it in this sentence

- Please write H2O2 using lower indices (L441), the same applies for other chemical formulas in the whole manuscript

- L461: mL - please unify, once you use ml, cm3 or mL

- where there any statistical analyses applied? One can see many effects on figures/tables which need to be explained in terms of their significance.

- L127: please correct 'ZnONps'

- Figure 2 caption is quite strange - first two sentences should re-phrased or changed. Again 'Bacillus" is not italized

- L253: please give full name of this stain (in italics) as it is mentioned for the first time

- L256: please write superoxide anion fomula using upper/lower indices

- L327: the reference [9] is cited here with name of author not a number

- L330-31: please use lower-indices for TiO2

- L345: there is no need to italize 'and'

- when writing of unit of liter unify the unit, sometime it is 'l' and sometimes 'L'

- L366-68: take care about indices in formulas

Author Response

Response to Reviewer 2 Comments

Thank you for your time and suggestions, which have helped us to improve our article, and for the overall positive opinion of the research objective, choice of methods, and analysis of the results.

Suggestion: Experiment is well designed and described allowing its repetition. I only miss statistical procedures as there are clear patterns which should be confirmed by statistical analysis (dose effects, differences between strains, differences to the control).

Authors: Thank you for this suggestion. We have added the results of the statistical analysis in Table 1. We have completed captions under the figures.  We also added conclusions in the discussion (L: 405-41; L:420-424).

Suggestion: I would also provide IAA values for each treatment for better comparison with data presented in discussion.

Authors: The IAA concentrations in the controls are given in Chapter2.4 L:203-205. In our opinion, the description of the results in this form is more appropriate.

Suggestion: Discussion is good using recent literature data, it ends with conclusions. However, conclusions need to be re-written taking into account the results of statistics and respond to the aim more in detail.

Authors: We supplemented the conclusions in the discussion L: 405-41; L:420-424.

Suggestion: The titles of subsections in methods should be written according to the Authors Guide

Authors: This was our mistake. We have improved the numbering in the material and methods chapter

Suggestion: what was the origin of stains used?

Authors: We have completed the description of the strains L:435-438

In the revised version of the manuscript we have taken into account all of the comments concerning editing – removed the double spaces, explained the abbreviations, and corrected the spelling errors. We have also changed some of the units so that they are consistent.

Thank you for your insightful analysis of our manuscript.